# Changes in Young Adults’ Perception of an Interspecific Hybrid Grape Juice Induced by the Addition of Acid or Sugar as Part of a Novel Diversification Strategy for the Grape Industry

**DOI:** 10.3390/foods14071170

**Published:** 2025-03-27

**Authors:** Georgia Lytra, Elie Maza, Julie Bornot, Olivier Geffroy, Christian Chervin

**Affiliations:** 1Univ. Bordeaux, INRAE, Bordeaux INP, Bordeaux Sciences Agro, UMR 1366, OENO, ISVV, 33882 Villenave d’Ornon, France; 2LRSV, Université de Toulouse, AgroToulouse, INP, BP 32607, 31326 Castanet-Tolosan, France; elie.maza@ensat.fr (E.M.); christian.chervin@toulouse-inp.fr (C.C.); 3LGC, Université de Toulouse, CNRS, INPT, UPS, 31326 Toulouse, France; julie.bornot@toulouse-inp.fr; 4PPGV, Université de Toulouse, Ecole d’Ingénieurs de PURPAN, 75 voie du TOEC, 31076 Toulouse, France; olivier.geffroy@purpan.fr

**Keywords:** interspecific hybrid grapes, viticulture, Seibel 5455, low-chemical input, fruit aroma

## Abstract

As an alternative to traditional red wine production during an economic crisis, we investigated the sensory perception and appreciation (liking) of young consumers for organic red grape juice from the Plantet cultivar (Seibel 5455) with or without the addition of sugar or organic acid. This was evaluated through four studies involving panels of young adults with an average age of 22 years. The goal was to determine how adding tartaric acid or hexoses (glucose and fructose) affected hedonic scores as well as gustatory and aromatic attributes. Surprisingly, adding acid to a must that experts deemed overly sweet (with 255 g/L of endogenous sugar in the control) did not improve liking scores. Similarly, the addition of hexoses did not enhance liking. Instead, natural grape juice, without any added sugar or acid, was the most preferred product among the young adult panelists. This preference was shared by a panel of enology students, who can be considered future key decision makers in the wine and juice industries, in the last study. As expected, the addition of hexoses led to a perception of ‘jam’ and ‘sweet’ attributes, while the addition of tartaric acid resulted in a more ‘acidic’ profile. Interestingly, the fruit attributes most associated with hexose addition were ‘apricot’ and ’strawberry’, whereas tartaric acid addition was most strongly associated with ‘raspberry’.

## 1. Introduction

A study by the Bureau of Statistics of the International Organization of Vine and Wine (OIV) shows that world consumption of red wine has decreased by 15% in the last fifteen years, representing a drop of 20 million hectoliters [1]. This decline in consumption, leading to a surplus, is particularly pronounced for red wines targeted at the lower segments of the market [2]. While the world consumption of rosé wine slightly increased over the same period, the total world consumption of rosé ‘only’ represents 20 million hectoliters/year. Thus, hoping to double the rosé consumption to compensate for the decrease in red wine consumption is pointless. To adjust to this situation, many growers are pulling up vines of red grape cultivars [3,4], sometimes replanting other cultivars. One potential alternative is to plant white grape cultivars, but the world consumption of white wine has been plateauing since 2017 [1]; another is the planting of interspecific hybrid vines, which is receiving increasing interest in the South of France in particular [5]. Hybrid vines are often associated with a reduction in production costs and a decrease in the need for chemical inputs; in some other countries, such as the USA, these hybrids are produced historically [6], and in other countries, like Switzerland, initiatives to convert part of the vineyards to low-chemical input viticulture are being promoted [7]. An advantage is that some interspecific grape hybrids, being fungus resistant, can be left on the vines until they reach the desired level of maturity while limiting the development of rot [8]. While red wine consumption may or may not resume its high levels in the coming decades, some producers are interested in valorizing red grapes as fruit juices, either sold as bioactive ingredients or used as sweeteners in various food products [9,10]. Replacing red wine, which is typically consumed with meals, with fruit juice is not straightforward. However, some researchers promote fruit juice consumption as a way to help meet the recommended five-a-day intake of fruits and vegetables [11]. However, the sugar/acid concentration of juices extracted from hybrid grape varieties can differ from those of *Vitis vinifera* or other fruit juices. Indeed, such hybrids can produce grapes with a very high titratable acidity at harvest (up to 15 g/L expressed as tartaric acid), which might require the addition of sugar to rebalance [12,13]. Moreover, sugar concentrations in hybrid juices are typically higher than in traditionally consumed fruit juices, such as apple or orange juice, in which sugar content seldom exceeds 16° Brix [14], which can justify the use of exogenous acid. To our knowledge, no specific work has been conducted on the sugar–acid balance of grape juices made from hybrid varieties, notably those produced in the northeast of the USA. Most of the research has been focused on the impact of the processes on Concord juice quality parameters or nutritive values [15,16]. Here, we are reporting the results of a series of tests to determine if young adults would prefer a grapevine hybrid juice with or without the addition of acid or sugar and if these additions would influence their perception towards certain fruit aromas. Young consumers were targeted, as they are important juice consumers [17].

## 2. Materials and Methods

### 2.1. Grapevine and Juice Process

The fruits were produced from 10-year-old red grapevines (cv. Plantet, also called Seibel 5455, an interspecific hybrid of several *Vitis* sp., undetermined lineage, marketed by Albert Seibel under the number 5455) grown on 3309C rootstock near the AgroToulouse school, Auzeville-Tolosane, France. The grapevines were sprayed against powdery and downy mildew, with only three treatments of combined copper and sulfur over the whole season. For each treatment, the dose applied was 270 g copper sulfate/ha (SOBB20800, Solabiol) and 1 kg micronized sulfur/ha (SOPUL400, Solabiol, SMB, Ecully, France). Each of the three treatments was applied at bud break, one week before full bloom and one week after full bloom, respectively.

A total of 800 kg of grapes were harvested in mid-September of 2024, with a yield of 1.1 kg fruit/vine. The clusters were immediately destemmed, and the berries were pressed at 150 bars in a vertical hydraulic press (Polsinelli, Isola del Liri, Italy, Ø80 TOP0035). After settling for 24 h, a 30 L sample of the juice was racked and heated to 80 °C to provide microbiological stability and was then poured immediately into bag-in-boxes, with the addition of 50 mg SO_2_/L and 50 mg potassium sorbate/L. The juices were kept at 5 °C until further analysis.

### 2.2. Biochemical Analyses of the Juices

The complete set of biochemical analyses performed on the juices can be found in Table 1. Spectrophotometric measures were performed with a visible spectrophotometer (Unico 2150, Dayton, OH, USA), and titratable acidity expressed as tartaric acid was determined by adding 0.1 M NaOH up to pH 7 [18]. The assays of glucose + fructose, lactic, malic, and tartaric acids were performed with an automatic biochemical analyzer, Mindray BS 230 (BioSentec, Portet sur Garonne, France), and the enzymatic reagent kits 072, 073, 070, and 045 (BioSentec, Portet sur Garonne, France), respectively. The malic acid, lactic acid, and acetic acid concentrations did not change in the juices supplemented with tartaric acid or hexoses).

### 2.3. Sensory Analyses

For the sensory analyses, eight attributes were chosen from a list generated in a preliminary meeting of 15 researchers in viticulture and enology from Montpellier and Toulouse. These attributes were the most frequently used to describe the control juice samples, and the experts were asked to focus on quality attributes that non-specialists would understand. A maximum of eight attributes was selected, as a recent article suggests that six to ten attributes seem to be optimal according to consumer studies [19].

The sensory analyses were performed over four subsequent studies, as described in detail below. Between-subjects experimental designs were used, and the questionnaire and data acquisition were performed with FIZZ Nomad v2.7 (BioSystemes, Couternon, France). All panelists answered a post-evaluation questionnaire with two questions to characterize their fruit juice consumption frequency and their juice preference (answers summarized in Table 2). This was followed by the Check-All-That-Apply (CATA) question, followed by the liking question, as it has recently been demonstrated that this order generates more discriminant analyses [19]. The random serving order of each of the various juices, all coded with three-digit numbers, was generated by FIZZ. Each panelist was served 20 mL of each juice at 16 °C in standard ISO wine glasses, which were either transparent or black, depending on the study (see details below). The panelists used their mobile phones to receive the questionnaires and submit their answers (via a QR code). In the CATA questionnaires, the attributes were randomly presented to each panelist. In all studies, at the start of each session, the panelists provided their informed consent and were told that their identities would remain anonymous. All panelist details are given in the following paragraphs and the associated Appendix A. A total of four different studies were conducted, with the results of one study defining the experimental design of the following one (which substance to add, doses).

Study 1:

A total of 103 panelists were recruited from the agricultural, physics, chemistry, and business students at the University of Toulouse in early October 2024 (see Table 2 for further details). We tested the impact of tartaric acid addition (Carlo Erba, analytical grade) on sensory description and appreciation. The tested doses were control (+0 g/L), dose 1 (+0.5 g/L), dose 2 (+1.5 g/L), dose 3 (+3 g/L), and dose 4 (+9 g/L). The sensory tests were performed using transparent glasses.

Study 2:

A total of 52 panelists were recruited from the agricultural students of AgroToulouse in early November 2024 (see Table 2 for further details). Studies 1 and 2 had 6% of the students in common. We tested the impact of the addition of two hexoses, glucose and fructose (Carlo Erba, analytical grade), on sensory description and appreciation. The tested doses were control (+0 g/L), dose 1 (+50 g hexoses/L), dose 2 (+100 g hexoses/L), and dose 3 (+150 g hexoses/L). Each sugar addition comprised 50% glucose and 50% fructose; thus, for example, ‘+ 50 g hexoses/L’ means +25 g glucose/L and +25 g fructose/L. The sensory tests were performed using transparent glasses.

Study 3:

A total of 49 panelists were recruited from the agricultural students at Bordeaux Sciences Agro in December 2024 (see Table 2 for further details). We tested the impact of both tartaric acid and hexose addition on sensory description and appreciation. The tested doses were hexoses + 150 (+150 g hexoses/L), hexoses + 50 (+50 g hexoses/L), control (+0 g/L), acid + 1.5 (+1.5 g tartaric acid/L), and acid + 9 (+9 g tartaric acid/L); see Section 4 regarding the dose choices. Hexoses were 50% glucose and 50% fructose, as described above. The sensory tests were performed using black glasses.

Study 4:

A total of 39 panelists were recruited from the enology students at the University of Toulouse in December 2024 (see Table 2 for further details). This study was conducted to investigate the perception of future key decision makers in the wine and juice industries and compare it with non-expert consumers (Study 3). We tested the impact of both tartaric acid and hexose addition on sensory description and appreciation, with identical doses to those used in Study 3. The sensory tests were performed with black glasses.

### 2.4. Data Analysis

All data were analyzed using R software (R Core Team, 2024, v.2022.07.1) and previously developed scripts [20]. All data sets and new versions of the scripts, which were generated and used in the four studies of the present article, are provided in Appendix A.

CATA citation scores were analyzed by correspondence analyses, and the dependence between attributes and juices was tested by chi-square tests of independence. The independence test was performed as a function of the number of panelists. Three hundred random draws were performed on the given total number of panelists, and boxplots of *p*-values were produced.

The liking scores were analyzed by two-way ANOVAs (panelist x product), and the multiple comparisons were then performed with Tukey’s tests at 0.05 (small letters shown on graphs). The probability of the observed acid/sugar effect, which is the corresponding *p*-value of the ANOVA, was drawn as a function of the number of panelists. Three hundred random draws were performed on the given total number of panelists, and boxplots of *p*-values were produced. The latter output is a new development from our earlier articles.

## 3. Results

The eight attributes selected for the CATA study were apricot, acidic, fruity, jam, plum, raspberry, sweet, and strawberry. Figure 1 shows the results of the first study, in which more than 100 young adults tasted different fruit juices with the addition of varying doses of tartaric acid.

A correspondence analysis clearly showed that, as expected, the highest dose of acid was associated with the ‘acidic’ attribute (Figure 1A). Interestingly, it was also associated with the ‘raspberry’ attribute. The less acidic juices were associated with ‘strawberry’ and ‘apricot’ and with ‘jam’ and sweet’ attributes. ‘Plum’ was associated with the addition of an intermediate acid dose. The percentage explained variance of the first factorial plan was higher than 99% (Figure 1A), and the dependence between attributes and juices was very strong for 27 panelists (Figure 1B); both these results confirm the robustness of our analyses.

The addition of tartaric acid at doses of 0.5 to 3 g/L did not generate significant differences in liking of the juices by the panelists (Figure 1C). However, the liking score diminished concomitantly as the dose increased, with the highest dose being 9 g/L, generating a significant drop in the liking score. The initial endogenous concentration of tartaric acid in the control juice was 4.2 g/L and the malic acid concentration was 4.4 g/L; these values are much lower than those usually found in hybrid grapes produced under cool climate vineyards [12]. Meanwhile, the sugar concentration was estimated to be 25 °Brix (Table 1). These analytical characteristics highlight the advanced maturity level of the grapes. Figure 1D shows that the addition of acid had a measurable significant effect on 36 panelists. Figure 1E shows slight differences in color as a function of acid addition: the highest dose of added acid resulted in a juice that was pinker than the control, as can clearly be seen in Table 1, with OD520 being increased by acid addition. The expert jury that validated the chosen attributes did not perceive any variability in astringency, probably because of the initial high sugar concentration; thus, the astringency attribute was not added to this study.

In the second study, we measured the effect of adding glucose and fructose on the description and appreciation of the juice by another panel of 52 young adults. The obtained figure (Figure 2A) shows that the taste resulting from the addition of hexoses was associated with the attributes ‘sweet’ and ‘jam’, and that of the control with ‘acidic’. While the association trends between juices and fruit aromas were less obvious, ‘apricot’ was also linked to the sweetest juices and ‘raspberry’ to the less sweet. It can be seen in Figure 2B that the better association of juices and attributes with a larger panel like 52 did not allow the 0.05 limit to be reached. In the set of ANOVA analyses, we observed that no significant differences were generated by the addition of sugar, and it can be seen that the control was preferred (Figure 2C). In addition, it is not possible to discern whether increasing the number of panelists would have had a significant effect on Figure 2D, as the slope towards the 0.05 limit is weak. The reason why we did not add higher sugar concentration in this second study is due to the fact that we nearly reached the solubility limit with +150 g/L in a juice that already contained more than 250 g/L.

In the third study, we investigated the effect of adding a sub-set of acid and sugar doses on the same sensory criteria (using CATA and ANOVA as above). The objective of this third study was to check if the impacts of acid or sugar addition, observed in separate studies, would or would not be similar when combined in a single study. The sensory analyses were performed using black glasses to avoid any bias toward associating the pinker juice with a raspberry attribute (Figure 1E). This study was conducted with a different group of panelists from the other studies in the university city of Bordeaux, instead of Toulouse. Figure 3A shows that ‘raspberry’ was still associated with acidic juices, even though the juices were tasted in black glasses. Furthermore, ‘strawberry’ and ‘apricot’ were associated with sweet juices, with a tendency for the strawberry attribute to be associated with sweeter juices than that of ‘apricot’. ‘Plum’ was intermediate and was markedly associated with ‘sweet’ or ‘acidic’. The addition of +50 g hexoses/L was not completely dissociated from the control. Figure 3B shows that there is a strong dependence between juices and attributes with 16 panelists or more. The results of the multiple comparisons shown in Figure 3C highlight that hexoses + 50 and control were appreciated with very similar liking scores (6.47 and 6.41, respectively) and that the only addition generating a significant difference in liking was the highest tartaric acid dose, leading to significant disliking relative to the control. Figure 4D shows that the acid or sugar addition generated a significant effect on liking with 24 panelists or more.

In the fourth study, we investigated the effect of adding a sub-set of acid and sugar doses, which was similar to Study 3. However, the session was run with a different panel, which was composed of enology students who were slightly more familiar with sensory analyses. The latter is evident in Table 2, with the highest % of panelists claiming to participate at least ‘sometimes’ or ‘often’ in sensory analyses being from Study 4. The results shown in Figure 4A are very similar to those shown in Figure 3A. The addition of acid to the juice was associated with the ‘acidic’ attribute, and the addition of sugar was associated with the ‘jam’ and ‘sweet’ attributes. As in earlier studies, the ‘raspberry’ attribute was associated with the highest acid dose. The ‘apricot’ attribute was associated with the sweetest juice, as was the ‘strawberry‘ attribute, which is weakly opposed on the vertical axis to the ‘plum’ attribute. Figure 3B shows a strong dependence between juices and attributes with 12 panelists or more. Figure 4C highlights similar trends in the multiple comparisons to Figure 3C. The only addition that generated a significant difference in liking was the highest tartaric acid dose, and once again, there was a tendency for the control juice to be preferred. Figure 3D shows that the acid or sugar addition generated a significant effect on liking with 15 panelists or more.

## 4. Discussion

### 4.1. Acid or Sugar Additions Do Not Improve Juice Liking

Surprisingly, the results of the first study show that none of the added acid doses increased appreciation of the juice (see liking scores in Figure 1C). It would have been interesting to test the addition of other acids, such as citric or lactic acids, but these can have an aromatic impact at a pH such as that of the juice [21]. The choice of tartaric acid in the present study was based on the fact that it is naturally present in grape juice in equal amounts to malic acid, but the latter is less stable because it is transformed into lactic acid by lactic bacteria [22].

Despite the high initial sugar concentration of the studied juice, the above results together with the initial hypothesis driving this work were the rationale behind adding more sugar. Young adults are known to be fans of soft drinks [23], and there is a positive correlation between sweet taste intensity and pleasantness as perceived by adults [24]. Due to the possibility that the optimal sugar concentration had not been reached in the control juice, we tested different juices to which we added sugars. We chose to add an equal amount of glucose and fructose because these are the two main hexoses of grape juice; they are usually present in equal concentrations as they result from sucrose cleavage by grape invertases [22]. The results of the liking scores (Figure 2C) show that none of the added sugar doses increased appreciation of the juice.

The number of panelists varied according to the studies, as young adults were recruited based on their availability in the different universities participating in the studies. Figure 1B,D, Figure 3B,D and Figure 4B,D showed that their number was sufficient to analyze the data with sufficient statistical strength. The only case where the panelist number could have been a limitation is in ‘study 2’ (Figure 2B,D), but as mentioned in the ‘study 2’ results paragraph, the slope towards the 0.05 limit is weak in Figure 2D; thus, the absence of significant difference is probably due to the fact that increasing sugar concentration does not generate any liking difference in a juice that has initially 250 g sugars/L. In both Figure 3B and Figure 4B, the *p*-value of the chi-square test of independence between attributes and juices went below the 0.05 limit because the only parameter inducing a significantly different perception is the acid addition. In Study 2, sugar addition does not create sufficiently big perception differences. And, indeed, in studies 3 and 4 in Figure 3A and Figure 4A, one can see that juices with varying sugar doses are quite close.

### 4.2. Impact of the Acid and Sugar Doses

To discuss the impact of acid or sugar on juice appreciation, we will first explain the choices of the various doses. With a control juice containing 4.3 g tartaric acid/L and 4.4 g malic acid/L, the addition of +0.5, +1.5, +3, and +9 g/L of tartaric acid corresponds to approximately +5%, 17%, 35%, and 103% of dicarboxylic acids. The highest acid dose corresponds to doubling the acidity of the juice, and the final pH was 2.43 (Table 1), close to 2.5, which is also the pH of Coca-Cola, a soft drink popular among young consumers [25]. However, in our case, the high acidity level led to a negative response from our young adult panelists.

Regarding sugars, with the control juice containing 255 g/L, the additions of +50, +100, and +150 g/L resulted in up to a 60% increase in the main hexoses compared to the control grape juice. The sweetest juice in our study was as sweet as a Tokay must [26]. The perception of sugars in sensory analyses is a complex issue, and variability in the sweet taste preferences of adults has been shown to be high [24]. Additionally, the sweetening power of fructose is approximately 2.1-fold that of glucose [27]; however, this did not affect our study, as we did not compare our hexose mix with other sweeteners.

The addition of sugar did not impact sensory appreciation as much as the addition of acid. Figure 2D indicates that increasing the panel size may not have had any significant effects, as the slope toward the 0.05 threshold is gradual and seems to approach an asymptote at around 0.2. However, this lack of effect of adding sugar probably depends on the initial sugar level of the beverage. While our results do not align with this finding, it is plausible that if our study had been conducted with less ripe hybrid grape juice—naturally lower in sugar and higher in acidity—the outcomes could have been different.

### 4.3. Acid or Sugar Addition Modifies the Perception of Fruity Attributes

It is interesting to note the association of the ‘raspberry’ attribute to the most acidic juices. In the first study in which transparent glasses were used, the ‘raspberry’ attribute was associated with the highest acid dose, an association that was thought to have been biased by the fact that acid addition made the juice pinker (Figure 1E), as assessors are known to be highly influenced by visual cues [28,29]. In the last two studies, however, despite black glasses being used, the more acidic juices were still associated with the ‘raspberry’ attribute, ruling out the explanation of a visual bias. An avenue to explore in further studies regarding this association is the possible influence of different acid concentrations on the release of aroma compounds, as has been shown for model wines [30].

Our results show that Plantet grape juices have aromas that are similar to apricot, strawberry, plum, and raspberry, depending on the level of sweetness or acidity of the juices; they can, therefore, be used as alternatives to pure juices or in juice blended with other fruit crops. These results also clearly show that fruity perceptions are influenced by the non-volatile matrix composition, such as sugar and acid content [30], or even by other non-volatile compounds, such as proanthocyanidic tannins [31], as previously demonstrated in model wine. Our study was restricted to the list of fruity attributes obtained in the preliminary session by the experts. These attributes produced very good sums of eigenvalues in the correspondence analyses (90.54% to 99.22%), resulting in two-dimensional graphs that were sound representations of the associations between juices and attributes. In further studies, however, it would be worth taking into account other fruity attributes mentioned in free-comment sessions with young adults [32].

Finally, it was surprising to find that the young adults had a low preference for grape juice, as revealed by the panel characterization questions (Table 2). This is perhaps due to the lower quantities of grape juice on the European and world markets as compared to apples and oranges (FAOSTAT data, https://www.fao.org/faostat/en/#home, accessed on 15 December 2024). The fact that the sugar/acid balance of grape juice highly differs from the other juices might also play an important role. Whether or not it is possible for the percentage of grape juice in the fruit juice market to increase remains unknown.

## 5. Conclusions

A liking score of over 6 on a 1-to-9 scale obtained for the juices in our study indicates strong consumer acceptance, highlighting their high potential in the young adult market. This finding could encourage winemakers to explore the production of juices using Plantet fruits or other interspecific hybrids.

Given current nutritional recommendations advocating for a reduction in daily sugar intake, future research should focus on evaluating the potential of these juices when made from grapes harvested at an earlier stage with lower sugar concentrations.

This study also demonstrates the effectiveness of specific methods and statistical tools in assessing the impact of acidity and sweetness adjustments to meet consumer preferences. However, to further characterize this product category, additional marketing studies are needed. These should compare various fruit juices and their pricing while also considering other influential factors, such as consumer perception of organic farming practices.

## Figures and Tables

**Figure 1 foods-14-01170-f001:**
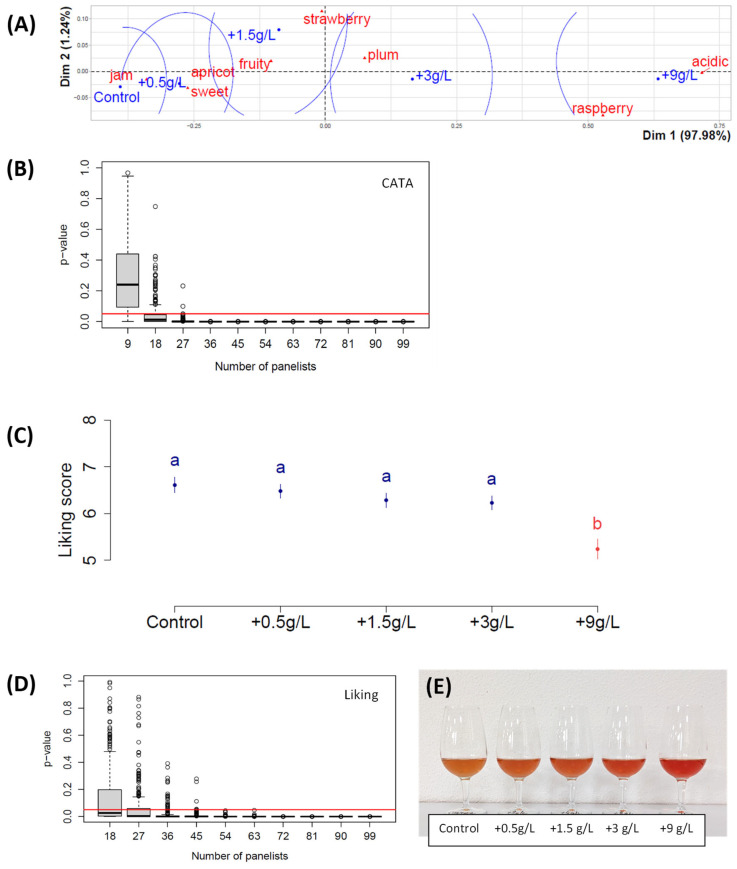
Results of Study 1 with Plantet grape juices and the addition of tartaric acid at various doses, expressed in g/L. (**A**) Correspondence analysis; (**B**) *p*-value of the chi-square test of independence between attributes and juices as a function of the panelist number, where the black line in the middle of the box is the median and the red line shows the 0.05 limit; (**C**) means of liking scores as a function of the juices, where error bars represent the standard error of the mean and little letters show significant differences calculated with Tukey’s test at 5% following ANOVA; (**D**) *p*-value of the absence of influence of the juice factor on the liking, in the ANOVA, as a function of the panelist number; n = 103 panelists; (**E**) color difference as a function of acid addition.

**Figure 2 foods-14-01170-f002:**
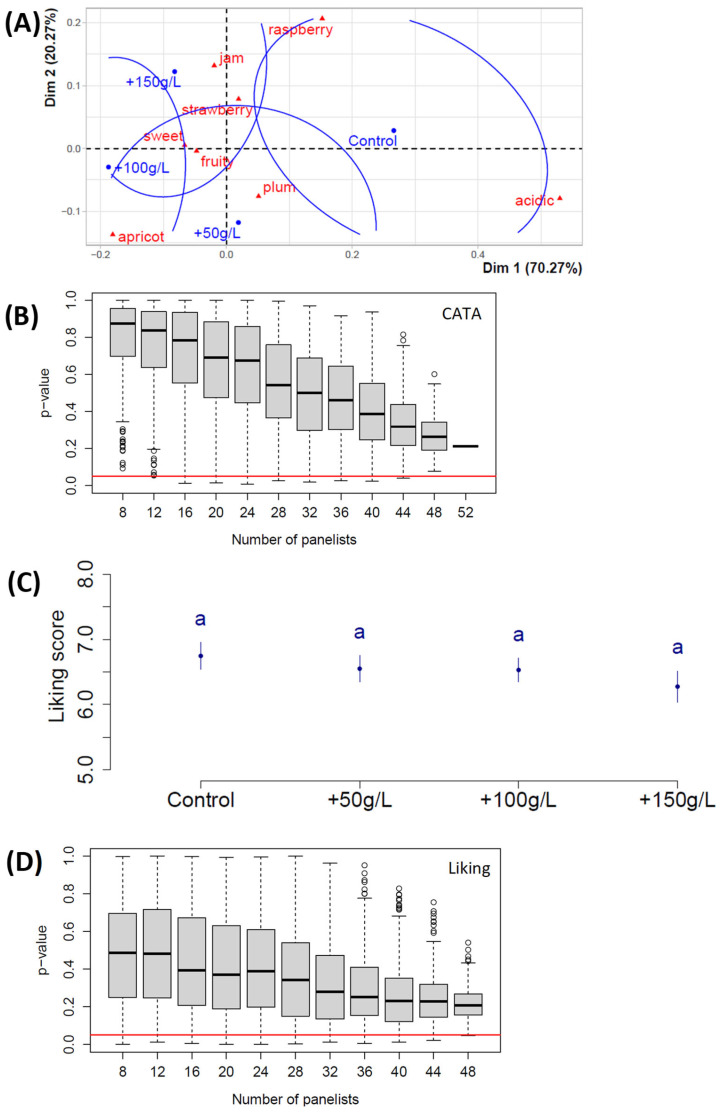
Results of Study 2 with Plantet grape juices and the addition of hexoses (50% glucose/50% fructose) at various doses, expressed in g/L. (**A**) Correspondence analysis; (**B**) *p*-value of the chi-square test of independence between attributes and juices as a function of the panelist number, where the black line in the middle of the box is the median and the red line shows the 0.05 limit; (**C**) means of liking scores as a function of the juices, where error bars represent the standard error of the mean and little letters show significant differences calculated with Tukey’s test at 5% following ANOVA; (**D**) *p*-value of the absence of influence of the juice factor on the liking, in the ANOVA, as a function of the panelist number; n = 52 panelists.

**Figure 3 foods-14-01170-f003:**
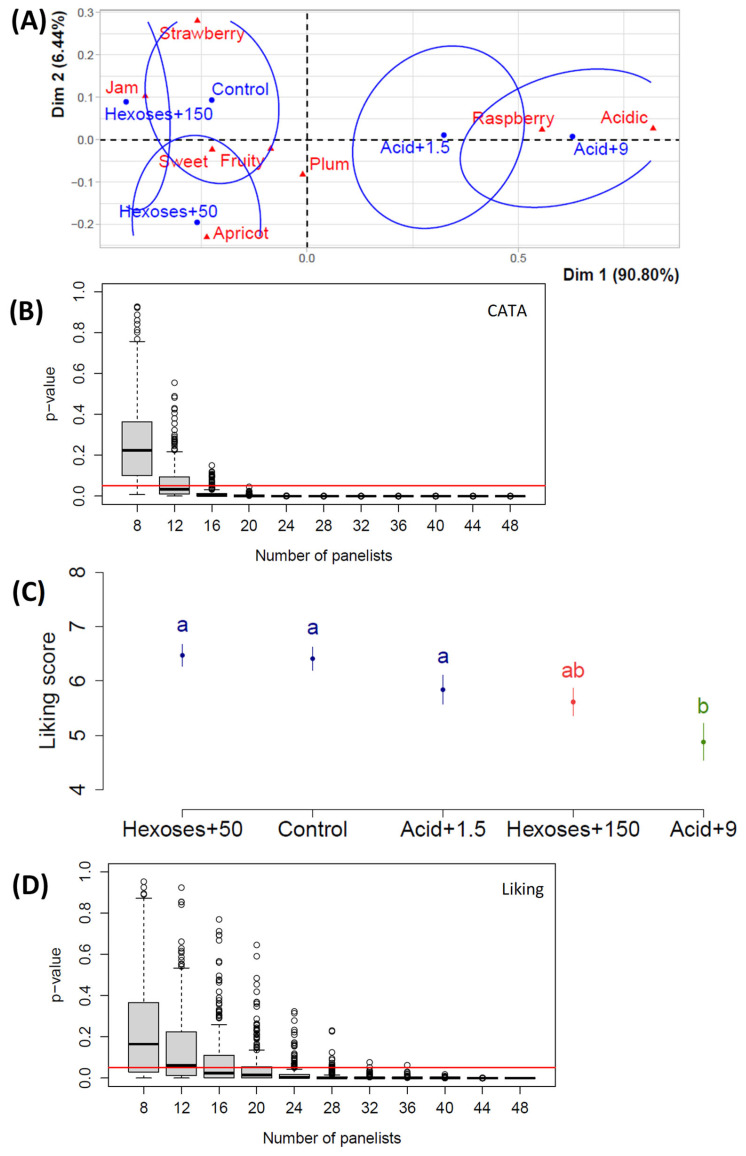
Results of Study 3 with Plantet grape juices and the addition of tartaric acid or hexoses at various doses, expressed in g/L. (**A**) Correspondence analysis; (**B**) *p*-value of the chi-square test of independence between attributes and juices as a function of the panelist number, where the black line in the middle of the box is the median and the red line shows the 0.05 limit; (**C**) means of liking scores as a function of the juices, where error bars represent the standard error of the mean and little letters show significant differences calculated with Tukey’s test at 5% following ANOVA; (**D**) *p*-value of the absence of influence of the juice factor on the liking, in the ANOVA, as a function of the panelist number; n = 49 panelists.

**Figure 4 foods-14-01170-f004:**
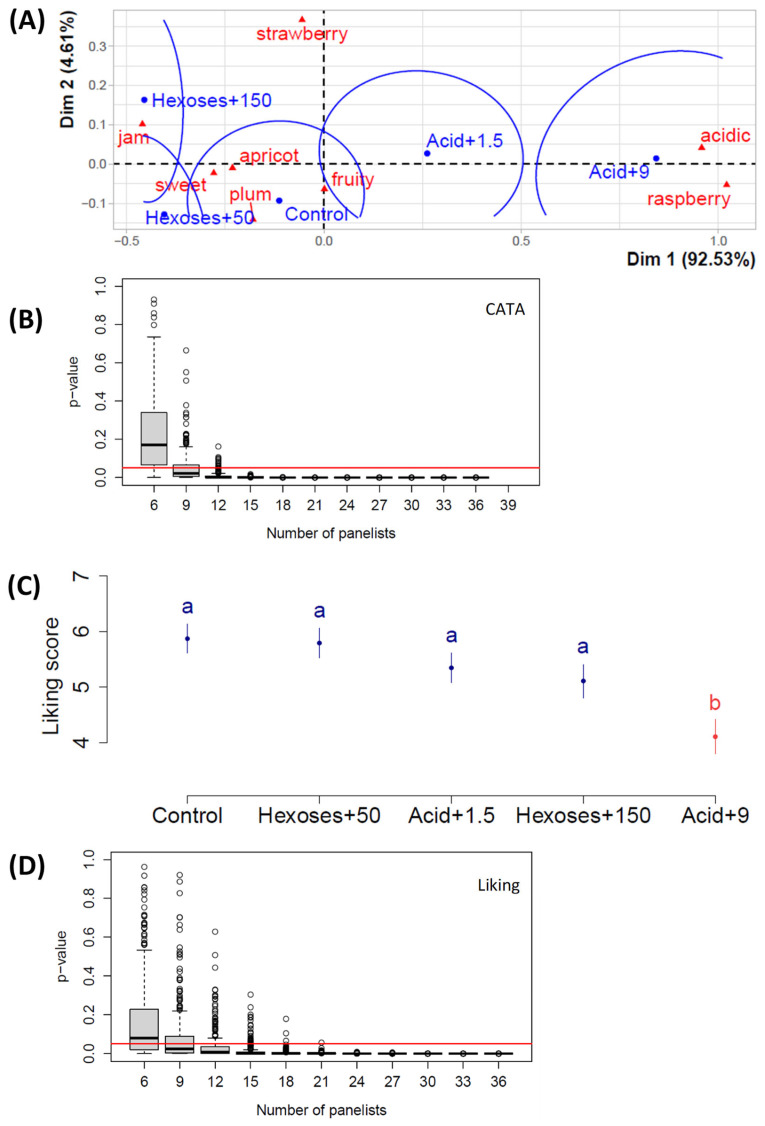
Results of Study 4 with Plantet grape juices and the addition of tartaric acid or hexoses at various doses, expressed in g/L. (**A**) Correspondence analysis; (**B**) *p*-value of the chi-square test of independence between attributes and juices as a function of the panelist number, where the black line in the middle of the box is the median and the red line shows the 0.05 limit; (**C**) means of liking scores as a function of the juices, where error bars represent the standard error of the mean and lowercase letters show significant differences calculated with Tukey’s test at 5% following ANOVA; (**D**) *p*-value of the absence of influence of the juice factor on the liking, in the ANOVA, as a function of the panelist number; n = 39 panelists.

**Table 1 foods-14-01170-t001:** Biochemical analyses of the juices. n = 4 ± standard error; color intensity = sum of the 3 ODs. Different little letters indicate significant differences at the 0.05 level (Tukey’s test) within each column. Note: The samples derived from the same juice showed no significant variation in their malic, lactic, and acetic acid contents.

	OD 420 nm	OD 520 nm	OD 620 nm	Color Intensity	pH	Titratable Acidity (g equiv. tartaric ac./L) *
control	0.526 ^a^ ± 0.21	0.33 ^a^ ± 0.21	0.104 ^a^ ± 0.21	0.96 ^a^	3.31 ^d^ ± 0.21	8.1 ^a^ ± 0.3
+0.5 g/L	0.529 ^a^ ± 0.21	0.361 ^a^ ± 0.21	0.108 ^a^ ± 0.21	0.998 ^a^	3.23 ^cd^ ± 0.19	8.6 ^a^ ± 0.3
+1.5 g/L	0.527 ^a^ ± 0.21	0.405 ^ab^ ± 0.21	0.097 ^a^ ± 0.21	1.029 ^ab^	3.07 ^bc^ ± 0.18	10.5 ^b^ ± 0.2
+3 g/L	0.548 ^a^ ± 0.21	0.459 ^b^ ± 0.21	0.099 ^a^ ± 0.21	1.106 ^b^	2.83 ^b^ ± 0.22	13.2 ^c^ ± 0.4
+9 g/L	0.549 ^a^ ± 0.21	0.546 ^c^ ± 0.21	0.147 ^a^ ± 0.21	1.242 ^c^	2.43 ^a^ ± 0.24	18.1 ^d^ ± 0.5
* the control juice contains approximately equal amounts of tartaric and malic acids (see below)						
		**Tartaric Acid (g/L)**				**Hexoses (g/L)**
control		4.28 ^a^ ± 0.30		control		254.4 ^a^ ± 2.6
+0.5 g/L		4.68 ^ab^ ± 0.26		+50 g/L		309.4 ^b^ ± 1.5
+1.5 g/L		5.35 ^b^ ± 0.31		+100 g/L		344.9 ^c^ ± 1.7
+3 g/L		7.32 ^c^ ± 0.42		+150 g/L		388.2 ^d^ ± 1.9
+9 g/L		14.28 ^d^ ± 0.82				
		**Malic Acid (g/L)**		**Lactic Acid (g/L)**		**Acetic Acid (g/L)**
control		4.42 ± 0.04		0.18 ± 0.04		0

**Table 2 foods-14-01170-t002:** Panel characterization; see panel locations and dates in the Materials and Methods.

	Study 1	Study 2	Study 3	Study 4
**Number of panelists**	103	52	49	39
**Age** (± SD)	21.1 ± 2.8	22.2 ± 1.8	22.2 ± 2.0	24.4 ± 3.9
**Gender**	W 52%; M 48%	W 65%; M 35%	W 53%; M 47%	W 59%; M 41%
**Freq. Cons. Juice ***				
Never	3.9	3.9	0.0	2.6
AFGpY	14.7	7.8	21.6	28.9
AFGpM	46.1	51.0	45.1	36.8
AFGpW	29.4	33.3	31.4	28.9
AFGpD	5.9	3.9	2.0	2.6
**Preferred Juice °°**				
Orange	23.5	23.5	31.4	39.5
Grape	14.7	9.8	9.8	5.3
Apple	46.1	60.8	39.2	34.2
Other	15.7	5.9	19.6	21.1
**Freq. Partic. Senso. A. ^++^**				
Never	79.4	51.0	45.1	23.7
Sometimes	20.6	45.1	45.1	55.3
Often	0.0	3.9	9.8	21.1

* Freq. Cons. Juice: the question was ‘At what frequency do you consume fruit juices?’; AFGpY = a few glasses per year, M = month, W = week, and D = day. °° Preferred Juice: the question was ‘What fruit juice do you prefer?’. ^++^ “Freq. Partic. Senso. A.: the question was ‘At what frequency do you participate to sensory analysis sessions?’.

## Data Availability

The original contributions presented in this study are included in the article/Appendix A. Further inquiries can be directed to the corresponding author.

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
