# Peer review of "Changes in Young Adults’ Perception of an Interspecific Hybrid Grape Juice Induced by the Addition of Acid or Sugar as Part of a Novel Diversification Strategy for the Grape Industry"

_foods, 2025, doi:10.3390/foods14071170_

Round 1
Reviewer 1 Report
Comments and Suggestions for Authors
The paper in question deals with particular topics of primary interest to winemakers.
I read the article with interest, even though I am not an expert in the field. I propose below some considerations that I hope can be useful for improving the article.
The first issue is actually a question: what is the relationship, in terms of replaceability/fungibility, between 'fruit juices' and red wine?
The second issue is also a question: is the supposed decline in red wine consumption, which initially inspired the work, generalized, or does it concern lower-quality products? In this second case, as some studies suggest, the problems for winemakers could be different.
I also believe that the work would benefit greatly if:
- the research questions of the article were clearly identified and expressed;
- more details were provided on the various groups/samples used for the survey (within the article), especially regarding their statistical significance.
Finally, the limitations of the paper should be expressed more clearly.
Author Response
The paper in question deals with particular topics of primary interest to winemakers.
I read the article with interest, even though I am not an expert in the field. I propose below some considerations that I hope can be useful for improving the article.
The first issue is actually a question: what is the relationship, in terms of replaceability/fungibility, between 'fruit juices' and red wine?
Good point. Our main consideration was that red grape juice could be marketed as an alternative to red wine, meaning that producers would need to compete for shares in the existing fruit juice market. However, the question of replacing red wine with red grape juice is indeed worth exploring. We have added the following sentence to the revised text: 'Replacing red wine, which is typically consumed with meals, with fruit juice is not straightforward. However, some researchers promote fruit juice consumption as a way to help meet the recommended five-a-day intake of fruits and vegetables (Benton and Young, 2018).' Red grape juice can also be a valuable alternative for food and soft drink pairing.
The second issue is also a question: is the supposed decline in red wine consumption, which initially inspired the work, generalized, or does it concern lower-quality products? In this second case, as some studies suggest, the problems for winemakers could be different.
Good point. The main concern relates to entry-level red wine. We have made the following changes in the manuscript: 'This decline, generating a surplus, is particularly marked for red wines targeted at the lower segments of the market (Del Rey & Loose, 2023).'
I also believe that the work would benefit greatly if:
- the research questions of the article were clearly identified and expressed;
Answer: we replaced the last sentence of the introduction to be more specific by: “We conducted a series of tests to determine if young adults would prefer a grapevine hybrid juice with or without the addition of acid or sugar, and if these additions would influence their perception towards certain fruit aromas.”
- more details were provided on the various groups/samples used for the survey (within the article), especially regarding their statistical significance.
Answer : Your remark has been taken into consideration and these data were analyzed using Tukey’s test.
Finally, the limitations of the paper should be expressed more clearly.
Answer : The manuscript was modified to discuss in more depth about the limitations of the study and future research to be conducted.
Reviewer 2 Report
Comments and Suggestions for Authors
This manuscript assesses how the addition of tartaric acid or hexose affects hedonic scores as well as taste and aroma attribute scores. The experimental design is reasonable, and the amount of data collected is sufficient. However, there are some problems in the details and experimental results of this manuscript. Therefore, major revision is required.
Major comments:
- The introductory section would benefit from a more comprehensive synthesis of current research progress in hybrid grape juice. Specifically, incorporating recent advancements in sugar-acid interaction mechanisms and critical gaps in dosage-response relationships would strengthen the rationale for this investigation.
- Authors chose 50,100,150g/L of sugar to add to the juice, and the results showed that the effect of added sugar on sensory preference was not significant. Please address the rationale for not exploring hypertonic solutions (>150 g/L).
- The experimental design should explicitly address the potential astringency modulation induced by tartaric acid-tannin complexation. Quantitative assessment of astringency perception is recommended to validate sensory outcomes.
- As for the purpose of experiment 3, please address the logic relationship with the first two experiments. By the way, the number of people used in your four studies is not consistent, and the reasons for choosing their numbers are not specified.
- Based on Fig 2 (B) p-value of the Chi-square test of independence between attributes and juice, the sugar contents have not relationship with panelists numbers. While, in Fig 3 (B) the result seems the contrary to Fig 2, please provide the explanation about the differences.
- For the description of Figure 2C and Figure 2D, it should be more specific to provide whether adding sugar or increasing the number of group members has a significant difference in some aspect. What do these blue curves in the four Correspondence analysis graphs in Fig 1-4 (A).
Comments on the Quality of English Language
The English could be improved to more clearly express the research.
Author Response
Comments and Suggestions for Authors
This manuscript assesses how the addition of tartaric acid or hexose affects hedonic scores as well as taste and aroma attribute scores. The experimental design is reasonable, and the amount of data collected is sufficient. However, there are some problems in the details and experimental results of this manuscript. Therefore, major revision is required.
Major comments:
- The introductory section would benefit from a more comprehensive synthesis of current research progress in hybrid grape juice. Specifically, incorporating recent advancements in sugar-acid interaction mechanisms and critical gaps in dosage-response relationships would strengthen the rationale for this investigation.
Answer, good point, the following information was included in the manuscript.
“To our knowledge no specific work has been conducted on sugar-acid balance of grape juices made from hybrid varieties notably those produced in the North-East of the USA. Most of the research have been focusing on the impact of the processes on Concord juice quality parameters or nutritive values (Dutra et al., 2021; Li & Padilla-Zakour, 2024).”
- Authors chose 50,100,150g/L of sugar to add to the juice, and the results showed that the effect of added sugar on sensory preference was not significant. Please address the rationale for not exploring hypertonic solutions (>150 g/L).
Answer: We added the following sentence at the end of the ‘study 2’ paragraph: “The reason why we did not add higher sugar concentration in this second study is due to the fact that we nearly reach the solubility limit with + 150g/L, in a juice that already contained more than 250g/L.”
- The experimental design should explicitly address the potential astringency modulation induced by tartaric acid-tannin complexation. Quantitative assessment of astringency perception is recommended to validate sensory outcomes.
Answer: Good point, we added the following sentence at the end of the ‘study 1’ paragraph: “The expert jury which validated the chosen attributes did not perceive any variability in astringency, probably because of the initial high sugar concentration, thus the astringency attribute was not added in this study.” Moreover, the consumers are not familiar with the ‘astringent’ term, they usually score it as ‘rough’. It can also be pointed out that tannins responsible for astringency are hydrophobic skin compound usually found at a low level of concentration in grape juice whose process involves a prefermentation skin removal and a maceration without ethanol, a solvent favoring tannins extraction.
- As for the purpose of experiment 3, please address the logic relationship with the first two experiments.
Answer: we added the following sentence at the beginning of the ‘study 3’ paragraph: “The objective of this third study was to check if the impacts of acid or sugar addition, observed in separate studies, would or would not be similar when combined in a single study”
By the way, the number of people used in your four studies is not consistent, and the reasons for choosing their numbers are not specified.
Answer: we added the following sentence at the end of the first paragraph in the Discussion chapter: “The number of panelists varied according to the studies, as young adults were recruited based on their availability in the different universities participating to the studies. The Figures 1B, 1D, 3B, 3D, 4B, 4D showed that their number was sufficient to analyze the data with a sufficient statistical strength.”
- Based on Fig 2 (B) p-value of the Chi-square test of independence between attributes and juice, the sugar contents have not relationship with panelists numbers. While, in Fig 3 (B) the result seems the contrary to Fig 2, please provide the explanation about the differences.
Answer: We added the following sentence at the end of the first paragraph in the Discussion chapter: “In both Figure 3B and 4B, the p-value of the Chi-square test of independence between attributes and juices went below the 0.05 limit, because the only parameter inducing a significantly different perception is the acid addition. In study 2, the sugar addition does not create sufficient big perception differences. And indeed, in study 3 and 4, in the Figure 3A and 4A, one can see that juices with varying sugar doses are quite close.”
- For the description of Figure 2C and Figure 2D, it should be more specific to provide whether adding sugar or increasing the number of group members has a significant difference in some aspect. What do these blue curves in the four Correspondence analysis graphs in Fig 1-4 (A).
Answer: We answered to the first part of the remark by adding a sentence in Discussion 4.1: « The only cases where the panelist number could have been a limitation is in ‘study 2’ (Fig. 2B, 2D), but as mentioned in the ‘study 2’ results paragraph, the slope towards the 0.05 limit is weak in Fig. 2D, thus the absence of significant difference is probably due to the fact that increasing sugar concentration does not generate any liking difference in a juice that has initially 250 g sugars/L. » The blue lines in all A) panels are confidence ellipses, classical elements in factorial analyses. They are described in Materials and Methods in the reference 19 and further discussed in this article.
Comments on the Quality of English Language
The English could be improved to more clearly express the research.
Answer: The English has been checked by a native English scientist.
Reviewer 3 Report
Comments and Suggestions for Authors
The article ‘Changes in young adults' perception of an interspecific hybrid grape juice induced by the addition of acid or sugar, as part of a novel diversification strategy for the grape industry ‘ presented for review is interesting. The abstract of the work provides a good introduction to its content. The introduction introduces the subject matter well. The authors clearly and comprehensively present the topics they want to address. Increased nutritional awareness and changes in habits and lifestyles affect the amount and type of food consumed. Food preferences may change with age, but younger consumers are more open to novelty. Acceptance of certain products by young consumers may result in them reaching for them more readily at a later time. Focusing on the preferences of this social group is therefore most advisable.
However, some issues need to be clarified or supplemented. The comments are included below.
- Why malic acid, lactic acid and acetic acid were tested only in the control sample
- The authors tested the possibility of improving flavor by adding sugar; however, the new trend nowadays is the possibility of producing juices with reduced sugar content. I suggest in future studies to take this into account. In the context of the current study, it is worth considering what interactions occurred between the different ingredients. The authors partially tried to clarify this issue.
- What do the authors think most influenced the fact that the addition of sugar did not increase the acceptability of the juice. Please explain this in the text. It is worth considering the effect of masking the sweet taste by the acids present in the product. For example, such an effect is observed in gastronomy when preparing tomato sauce, for example.
- In the reviewer's opinion, the authors should rely more on their own results in the conclusion. I find the citation of the literature in this chapter strongly questionable. The summary should be rewritten.
Author Response
Comments and Suggestions for Authors
The article ‘Changes in young adults' perception of an interspecific hybrid grape juice induced by the addition of acid or sugar, as part of a novel diversification strategy for the grape industry ‘ presented for review is interesting. The abstract of the work provides a good introduction to its content. The introduction introduces the subject matter well. The authors clearly and comprehensively present the topics they want to address. Increased nutritional awareness and changes in habits and lifestyles affect the amount and type of food consumed. Food preferences may change with age, but younger consumers are more open to novelty. Acceptance of certain products by young consumers may result in them reaching for them more readily at a later time. Focusing on the preferences of this social group is therefore most advisable.
However, some issues need to be clarified or supplemented. The comments are included below.
- Why malic acid, lactic acid and acetic acid were tested only in the control sample
Answer: We added the following sentence in the Material and Methods: “The malic acid, lactic acid and acetic acid concentrations did not change in the juices supplemented with tartaric acid or hexoses (unshown data).”
- The authors tested the possibility of improving flavor by adding sugar; however, the new trend nowadays is the possibility of producing juices with reduced sugar content. I suggest in future studies to take this into account. In the context of the current study, it is worth considering what interactions occurred between the different ingredients. The authors partially tried to clarify this issue.
Answer: We added the following sentence in the Conclusion chapter: “Given current nutritional recommendations advocating for a reduction in daily sugar intake, future research should focus on evaluating the potential of these juices when made from grapes harvested at an earlier stage, with lower sugar concentrations.”
- What do the authors think most influenced the fact that the addition of sugar did not increase the acceptability of the juice. Please explain this in the text. It is worth considering the effect of masking the sweet taste by the acids present in the product. For example, such an effect is observed in gastronomy when preparing tomato sauce, for example.
Answer: Indeed, the addition of acid can counterbalance the sweet taste. However, as explained in section 4.1, acid or sugar additions do not improve juice liking. The absence of a significant difference due to sugar addition in studies 2, 3, and 4 is probably because increasing the sugar concentration does not affect liking in a juice that initially contains 250 g of sugar per liter.
- In the reviewer's opinion, the authors should rely more on their own results in the conclusion. I find the citation of the literature in this chapter strongly questionable.
Answer: This paragraph has been modified accordingly.
The summary should be rewritten.
Thank you for your feedback. I have revised the summary to improve the English.